

# Identification of a novel intermittent hypoxia-related prognostic lncRNA signature and the ceRNA of lncRNA GSEC/miR-873-3p/EGLN3 regulatory axis in lung adenocarcinoma

Peijun Liu[1,*], Long Zhou[2,*], Hao Chen[1], Yang He[1], Guangcai Li[2] and Ke Hu[1]

[1] Department of Respiratory and Critical Care Medicine, Renmin Hospital of Wuhan University, Wuhan, Hubei, China

[2] Department of Respiratory and Critical Care Medicine, The Central Hospital of Enshi Tujia and Miao Autonomous Prefecture, Enshi Clinical College of Wuhan University, Enshi, Hubei, China

[*] These authors contributed equally to this work.

Corresponding authors
Guangcai Li, 44830705@qq.com
Ke Hu, huke-rmhospital@163.com

## ABSTRACT

**Background**. Lung adenocarcinoma (LUAD) is still the most prevalent type of respiratory cancer. Intermittent hypoxia can increase the mortality and morbidity associated with lung cancer. Long non-coding RNAs (lncRNAs) are crucial in lung adenocarcinoma. However, the effects of intermittent hypoxia-related long non-coding RNAs (IHRLs) on lung adenocarcinoma are still unknown.

**Method**. In the current research, eight IHRLs were selected to create a prognostic model. The risk score of the prognostic model was evaluated using multivariate and univariate analyses, and its accuracy and reliability were validated using a nomogram and ROC. Additionally, we investigated the relationships between IHRLs and the immune microenvironment.

**Result**. Our analysis identified GSEC, AC099850.3, and AL391001.1 as risk lncRNAs, while AC010615.2, AC010654.1, AL513550.1, LINC00996, and LINC01150 were categorized as protective lncRNAs. We observed variances in the expression of seven immune cells and 15 immune-correlated pathways between the two risk groups. Furthermore, our results confirmed the ceRNA network associated with the intermittent hypoxia-related lncRNA GSEC/miR-873-3p/EGLN3 regulatory pathway. GSEC showed pronounced expression in lung adenocarcinoma tissues and specific cell lines, and its inhibition resulted in reduced proliferation and migration in A549 and PC9 cells. Intriguingly, GSEC manifested oncogenic properties by sponging miR-873-3p and demonstrated a tendency to modulate EGLN3 expression favorably.

**Conclusion**. GSEC acts as an oncogenic lncRNA by interacting with miR-873-3p, modulating EGLN3 expression. This observation underscores the potential of GSEC as a diagnostic and therapeutic target for LUAD.

## INTRODUCTION

Among the most prevalent malignancies, lung adenocarcinoma (LUAD) has a higher incidence in the population (*Sung et al., 2021*). Due to the difficulty of early identification and therapy, LUAD remains one of the tumors with the poorest prognosis (*Thai et al., 2021*). Despite the discovery of various biomarkers and diagnostic equipment, there are still numerous gaps in the accurate diagnosis and therapy of LUAD (*Liu et al., 2021*; *Yu et al., 2021*). Intermittent hypoxia (IH) is a condition marked by intermittent oxygen deprivation, which may result in oxidative stress and inflammation (*Shobatake et al., 2022*). The primary trigger of intermittent hypoxia is obstructive sleep apnea (OSAS), which causes a combination of apnea and hypopnea due to a decrease in the size of the pharyngeal cavity. At the same time, the patient is asleep (*Labarca et al., 2020*). There has been less investigation into the association between intermittent hypoxia and lung cancer in previous studies. The severity of OSAS is associated with a relationship between OSAS and lung cancer, a molecular pathway involved in hypoxia-induced lung cancer progression (*Li et al., 2017*). Several studies show an association between intermittent hypoxia and cancer, which promotes carcinogenesis and progression through oxidative stress and inflammatory translation (*Navarrete-Opazo & Mitchell, 2014*; *Mateika et al., 2015*; *Almendros & Gozal, 2018*). Intermittent hypoxia in lung adenocarcinoma has been associated with a more aggressive tumor phenotype, increased metastatic potential, and resistance to therapy. It promotes the survival and proliferation of cancer cells, stimulates angiogenesis, and enhances the ability of tumor cells to invade and metastasize to distant sites. Furthermore, intermittent hypoxia has been shown to contribute to therapy resistance and poor patient outcomes.

Long non-coding RNAs (lncRNAs) are a class of RNA molecules that do not code for proteins but play critical regulatory roles in gene expression. They have been increasingly recognized as key players in various biological processes, including cancer development and progression (*Peng, Koirala & Mo, 2017*). In lung adenocarcinoma, aberrant expression and dysregulation of lncRNAs have been observed. These lncRNAs can act as oncogenes or tumor suppressors, influencing tumor growth, invasion, metastasis, and drug resistance. They can interact with other molecules, such as DNA, RNA, and proteins, to modulate gene expression and cellular processes involved in cancer development. Recent research has shown that lncRNAs regulated the early development of LUAD in multiple ways, including through different signalling pathways (*Song et al., 2021*). Downregulated DGCR5 expression was powerfully associated with smaller tumor size, indicating LUAD patients have a higher survival rate (*Dong et al., 2018*). Furthermore, the risk model of 12 ferroptosis-related lncRNAs has significant prognostic value for LUAD and may be ferroptosis-related therapeutic targets in the clinic (*Lu et al., 2021*). The Lactate Metabolism-Related lncRNA signature has also been related to immune cell infiltration and the immune checkpoint blocker CTLA-4 (*Mai et al., 2022*). Patients diagnosed with LUAD can have their prognosis accurately predicted using lncRNAs associated with cuproptosis (*Wang et al., 2022*). The above results indicate that lncRNAs may also play a role in the immune microenvironment of LUAD.

In recent years, there has been growing interest in understanding the interplay between lncRNAs and intermittent hypoxia in lung adenocarcinoma. Emerging evidence suggests that specific lncRNAs are responsive to hypoxia and are differentially expressed under low oxygen conditions. These hypoxia-responsive lncRNAs can modulate gene expression patterns associated with tumor growth, metastasis, and therapy resistance. They can act as mediators or regulators of hypoxia-induced signalling pathways, influencing the cellular response to intermittent hypoxia. Therefore, we investigate the association between intermittent hypoxia-related long non-coding RNAs (IHRLs) and LUAD.

In this work, we present a series of experiments to investigate the expression levels of IHRLs and their association with the immune microenvironment in LUAD. The potential regulatory mechanisms, including target genes and miRNAs, were further investigated. The results of this study will be helpful in the investigation of potential prognostic biomarkers associated with LUAD.

## MATERIALS AND METHODS

### Data collection and processing
The Cancer Genome Atlas (TCGA) data (https://portal.gdc.cancer.gov/) were used to obtain the transcriptional data and clinical characteristics of LUAD patients; patients with incomplete survival data were excluded. The clinical data was constructed utilizing Perl programming language (https://www.perl.org/version/5.32.1). The downloaded data is a single FPKM file for each sample. We use Perl language to merge all samples into a matrix for subsequent analysis. All the data applied in this research were freely accessible in a public database. There was no need for approval from an ethics commission.

### Identification of intermittent hypoxia-related genes (IHRGs)
Thirty-nine IHRGs were obtained from the GeneCards (https://www.genecards.org/) database according to the screening criterion: Relevance score >10. Thirty-two mRNAs were finally selected as differentially expressed genes (DEGs) by R software (version 4.3; *R Core Team, 2023*). The 32 DEGs were then sorted into a PPI network by STRING (https://string-db.org/), and the network was visualized using Cytoscape 3.4.0 software.

### Functional enrichment analysis of intermittent hypoxia-related DEGs
The enrichment analyses were performed by Kyoto Encyclopedia of Genes and Genomes (KEGG) (http://www.genome.jp/kegg/) and gene ontology (GO) (http://www.geneontology.org/). The biological characteristics of intermittent hypoxia-related DEGs were identified using these two databases.

### Construction of intermittent hypoxia-related lncRNAs prognosis model
Bioinformatics was used to identify lncRNAs DEGs associated with intermittent hypoxia. Patients ($n = 504$) were randomly allocated to training and validation groups in a 1:1 ratio. IHRLs were screened using univariate Cox regression analysis, the LASSO Cox algorithm, and multivariate analysis. Then, a prognostic model was established using the regression coefficient ($\beta$), the risk score $= \text{Expression}_1 * \beta_1 + \text{Expression}_2 * \beta_2 + \ldots + \text{Expression}_n * \beta_n$.

Then, we assessed the risk scores and clinical characteristics of LUAD using univariate and multivariate Cox regressions and ROC curves assessment.

## LncRNA risk score model correlation with tumor microenvironment infiltration

Based on gene expression profiles, CIBERSORT was used to calculate immune cell composition. The association between the risk model and LUAD immune infiltration was investigated utilizing R software. Furthermore, we apply the cor test function to calculate the relationship between risk score and immune cells.

## Construction of competing endogenous RNA axis

The Mircode database (http://www.mircode.org/) investigated the relationship between miRNAs and lncRNAs. The TargetScan database (https://www.targetscan.org/) was used to identify the downstream target mRNAs of miRNAs. The GEPIA database (http://gepia.cancer-pku.cn/) was used to analyze the expression of the downstream target mRNAs.

## Cell culture and cell transfection

LUAD cell lines A549 and PC9 were provided by Procell Life Science & Technology Co,. Ltd (Wuhan, China). The LUAD cells were cultured in DMEM medium (Gibco, Billings, MT, USA) with 10% FBS and maintained at 37 degrees Celsius in 5% $CO_2$ in an incubator. Si-GSEC, miR-873-3p inhibitor, and miR-873-3p mimics were synthesized by GenePharma (Shanghai, China). The LUAD Cells were plated in 96-well plates and transfected with plasmid by Lipo3000 (Invitrogen, Carlsbad, CA, USA). These cells were incubated at 37 degrees Celsius for 48 h and harvested for the subsequent experiments.

## Cell proliferation assay

For cell viability experiments, HBE, A549, PC9, and NCI-H1299 cells were seeded in 96-well plates and incubated for 24 h at 37 degrees Celsius and 5% CO2 in the incubator. After treating with the CCK-8 solution for 2 h, we measured the absorbance at 450 nm using a PerkinElmer EnSight (Waltham, MA, USA)

## Wound healing assay

The migration of cells was quantified using a wound-healing assay. After incubating the cells in 6-well dishes at 37 degrees Celsius, a line was drawn with a 200ul pipette tip. Images were captured at 0 and 24 h post-injury, and ten distinct sites were randomly selected for annotation and quantification.

## Colony formation assays

LUAD cells (A549 and PC9) were seeded into 6-well dishes, transfected the following day, and incubated for approximately two weeks. After two weeks, the cells were extracted, washed three times with PBS, and dyed with 4% paraformaldehyde and crystal violet solutions, respectively. Finally, the data is stored by photographing it with the camera for subsequent research.

**Table 1  Primer list.**

| Gene | Primers |
|---|---|
| GSEC | Forward: 5′-GAGTTCATTTGCTCTCTCTGGCAC-3′<br>Reverse: 5′-AAGAGGAGGCCTGATGGGGATA-3′ |
| EGLN3 | Forward: 5′-TCAAGGAGAGGTCTAAGGCAA-3<br>Reverse: 5′-ATGCAGGTGATGCAGCGA-3′ |
| GADPH | Forward: 5′-CCCCTTCATTGACCTCAACTACAT-3′<br>Reverse: 5′-CGCTCCTGGAAGATGGTGA-3′ |
| U6 | Forward: 5′-GCTTCGGCAGCACATATACTAAAAT-3′<br>Reverse: 5′-CGCTTCACGAATTTGCGTGTCAT-3′ |
| miR-873-3p | Forward: 5′-TTTGTGTGCATTTGCAGGAACT-3′<br>Reverse: 5′-GAAGATTTGTGGGTGTTCCCG-3′ |

## Dual-luciferase assay

The sequences of GSEC (GSEC wt and GSEC mut) and EGLN3 (EGLN3 wt and EGLN3 mut) containing the homologous binding sites of miR-873-3p were amplified and uniformly inserted into the vector pGL3 (Promega, Madison, WI, USA). The miR-873-3p mimics were then co-transfected with GSEC wt, GSEC mut, EGLN3 wt, or EGLN3 mut using Lipo3000. The DualLuciferase Reporter Assay Kit was used to measure luciferase activity.

## RNA immunoprecipitation assay

The RNA immunoprecipitation (RIP) experiment was conducted with the EZ-Magna RIP Reagent (Millipore, USA). The LUAD cells were lysed with RIP buffer and incubated with Anti-Ago2 or negative control (Anti-IgG). RT-qPCR identified the extracted RNAs after treatment with immunoprecipitated RNA and Proteinase K (Roche).

## Western blot

RIPA assay lysis buffer (RIPA: PMSF =100:1) was used to extract the protein of LUAD cells. After measuring all densities, the prepared proteins (30 µg/lane) were separated by SDS-PAGE and transferred into pre-cut PVDF membranes. After blocking in skimmed experimental milk (5%) for two h at 37 degrees Celsius, the targeted membranes were incubated with preconfigured primary anti-EGLN3 (ab30782; 1:1000; Abcam; Cambridge, UK) and $\beta$-actin (ab207327; 1:2000; Abcam; Cambridge, UK) overnight at 4 °C. The following day, the incubation with the associated secondary antibody was applied for 2 h before getting washed with PBS solution. Immunoreactive bands were displayed with the ECL Chemiluminescence Kit (G2020; Service) and visualization of the image capture system (ChemiDoc MP; Bio-Rad, Hercules, CA, USA).

## RT- qPCR analysis

Trizol (Beyotime, China) was used to extract RNA from cells. The SuperScript VILO cDNA Reagent (Invitrogen, Waltham, MA, USA) was used to convert RNA into complementary DNA (cDNA). SYBR Green qPCR Master Mix (Applied Biosystems, Waltham, MA, USA) was adopted to detect the quantitative PCR from the $2^{-\Delta\Delta Ct}$ method (*Livak & Schmittgen, 2001*). The RT-qPCR primer sequences are listed in Table 1.

## Statistical analysis

Statistical analysis was performed with the SPSS 25.0 software (SPSS Inc., Chicago, IL, USA), R software (version 3.5.1; *R Core Team, 2018*), and Graphpad Prism version 8 (Graphpad, San Diego, CA, USA). Univariate COX and multivariable Cox regression were conducted to evaluate the molecular clusters and classifier using the clinical signatures as a concomitant variable. One-way analysis of variance (ANOVA) and paired samples *t*-test are utilized to assess group differences, whereas Pearson's correlation test (was applied to examine correlations. Survival analysis was conducted using the Kaplan–Meier (KM) method, and a log-rank test was used for comparison. Each experiment was conducted independently and three times. It was deemed statistically significant if $P < 0.05$.

## RESULTS

### Identification of the gene expression related to intermittent hypoxia in LUAD

Figure 1 depicts the flowchart of the entire research. The expression of 39 genes related to intermittent hypoxia was evaluated, and 32 genes were identified as different expression genes (DEGs) as $^{*}p < 0.05$, $^{**}p < 0.01$, and $^{***}p < 0.001$. Compared with normal tissues, 19 genes (EGLN3, HILPDA, HMBS, HYOU1, BNIP3, *etc.*) were enriched, while 13 genes (SLC4A1, IL6, EPAS1, ACE, HMOX1, *etc.*) were decreased (Fig. 2A). PPI network was used to illustrate the interaction among 32 DEGs (Fig. 2B). HIF1A, VEGFA, and TP53 are identified as hub genes in the network (Fig. 2C).

### KEGG and GO enrichment analysis

KEGG and GO databases were performed to analyze the biological function of 32 DEGs. GO analysis showed that these genes in biological processes (BP) were involved in response to hypoxia", "response to chemical stress", and "neuron death". Alterations in cell component (CC) were engaged in "basal plasma membrane", "basal part of cell", and "membrane raft". And these genes in molecular function (MF) were involved in "carboxylic acid binding", "oxidoreductase activity," and "cytokine receptor binding" (Fig. 3A). EGLN3, SLC2A1, CA9 are the top three genes associated with GO enrichment (Fig. 3B). KEGG analysis showed that these genes were involved in the "HIF-1 signalling pathway", "AGE-RAGE signalling pathway", "IL-17 signalling pathway", and"ferroptosis (Figs. 3C–3D).

### Identification of IHRLs and co-expression network construction

Firstly, we classified the included cases ($n = 504$) into training and validation cohorts at a 1:1 ratio ($n = 252, 252$). 821 lncRNAs related to 32 DEGs were screened using the Pearson correlation method ($|R|>0.4$ and $p < 0.001$). PCA analysis was performed on the data based on all LncRNAs (Fig. 4A). Univariate Cox regression analysis and LASSO Cox algorithm were performed (Figs. 4B–4C). Finally, eight lncRNAs, including GSEC, AC099850.3, AL391001.1, AC010615.2, AC010654.1, AL513550.1, LINC00996, and LINC01150 were selected to construct the risk model, and the *p*-value of the risk model $=5.0106e-12$ (Fig. 4D). The co-expression network was shown in Fig. 4E. Among these 8 IHRLs, AC010615.2, AC010654.1, AL513550.1, LINC00996, and LINC01150 were identified as five protective

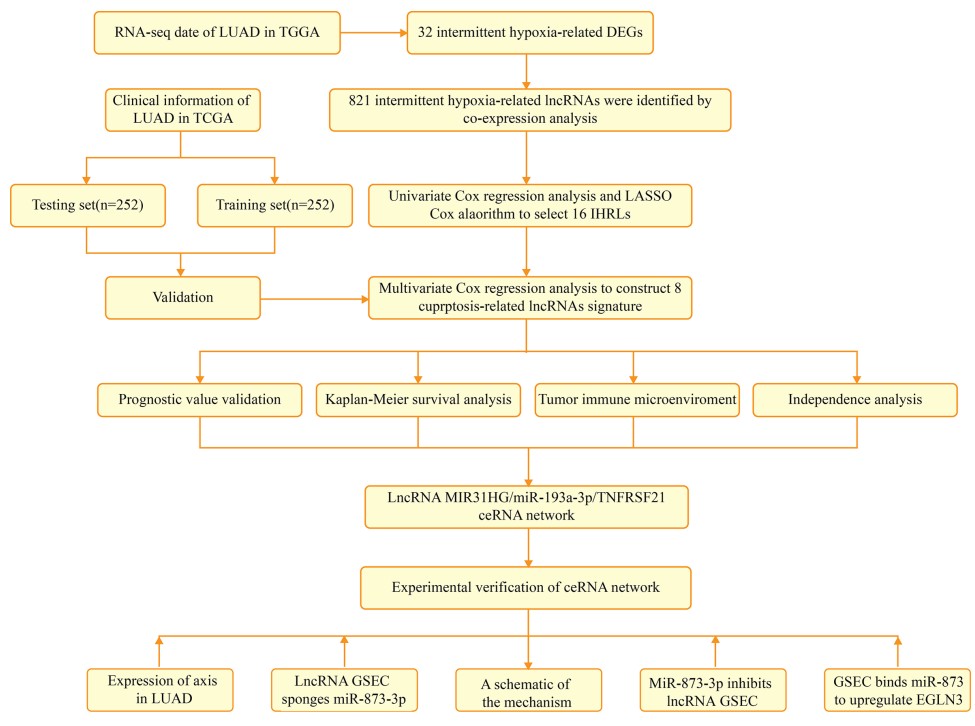

**Figure 1** The flowchart of the entire research.

lncRNAs. In contrast, GSEC, AC099850.3, and AL391001.1 were identified as three risk lncRNAs (Fig. 4F). Furthermore, Fig. 4G shows that the risk model *versus* all LncRNAs can effectively differentiate the two risk groups of patients.

## Construction of predictive risk model in LUAD patients

The risk score was calculated according to our formula as follows: Risk Score $= (0.2276 *$ Expression $_{GSEC}) + (-0.8246 *$ Expression $_{AC010615.2}) + (0.0707 *$ Expression $_{AC099850.3})$ $+ (0.5249 *$ Expression $_{AL391001.1}) + (-0.4247 *$ Expression $_{AC010654.1}) + (-0.5116$ $*$ Expression $_{LINC01150}) + (-0.6743 *$ Expression $_{LINC00996}) + (-0.5404 *$ Expression $_{AL513550.1})$. We divided all patients into high-risk or low-risk groups based on the median risk score. Figs. 5A–5B showed the expression heat map of these 8 IHRLs. Figs. 5C–5F shows the survival status of patients with different risks, and the results showed that patients in the high-risk group had significantly shorter overall survival (OS) ($P <0.05$) (Figs. 5G–5H).

## Prognosis value of model lncRNAs in LUAD

To determine how the risk score of the prognostic model affects survival, we performed univariate and multivariate Cox proportional hazards analyses. Our risk score of this prognostic model predicted LUAD OS independently in these investigations (Figs. 6A–6B). Risk scores were more specific and sensitive than other clinical features (AUC =0.782) (Fig. 6C).

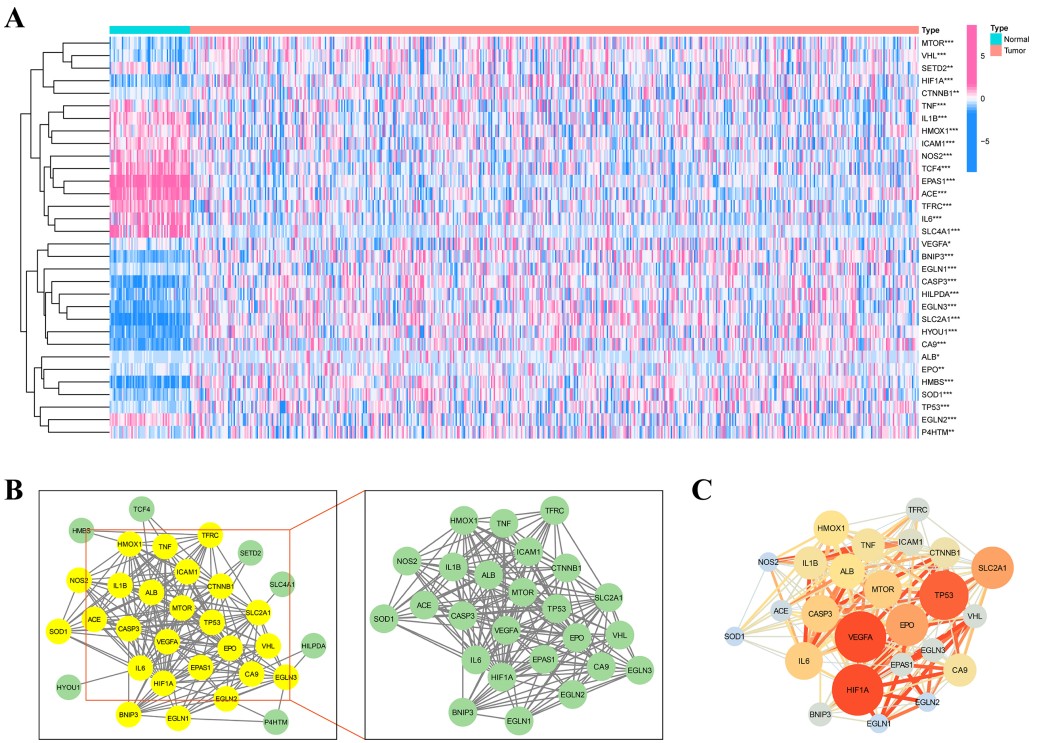

**Figure 2** **Identification of the intermittent hypoxia-related genes (IHRGs) expression and biological functional enrichment research.** (A) Heatmap of the 32 different expression genes (DEG) in LUAD and normal tissues. (B) PPI network of 32 IHRGs. (C) HIF1A, VEGFA, and TP53 are identified as hub genes in the network. $^*p < 0.05$; $^{**}p < 0.01$; $^{***}p < 0.001$.

## Correlation analysis of risk score and tumor microenvironment infiltration

We divided LUAD samples into low-risk and high-risk groups based on the median prognostic risk level. The expression of different immune cell infiltration results is illustrated in Fig. 7A. The enrichment analysis of oncogenic gene sets in the GSEA revealed these sets were mainly enriched in "RPS14 DN.V1 UP", "CSR LATE UP.V1 DN", "KRAS.KIDNEY UPV1 UP", "CYCLIN D1 KE .V1 DN", "AKT UP V1 DN", "AKT UP.V1 DN" (Fig. 7B). The relationship between risk score and infiltration of immune cells was described in Figs. 7C–7K, showing that T cells CD4 memory activated Macrophages positively correlated with a risk score. Meanwhile, there was a negative correlation between Dendritic cells resting, Mast cells resting, Monocytes, T cells CD4 memory resting, and T cells regulatory. In summary, these results highlight the immunomodulatory effects of the risk model.

## Construction of a ceRNA axis of lncRNA GSEC/miR-873-3p/EGLN3

We constructed a lncRNA-miRNA-mRNA regulatory axis to analyze the molecular mechanism of IHRLs. Using the Mircode database, eight miRNAs(miR-101-3p, miR-1185-5p, and miR-3679-5p, miR-873-3p, miR-134-5p, miR-3118, miR-296-5p, miR-4677-3p) were predicated as possible targets of GSEC (Fig. 8A). Compared with normal lung tissue, GSEC is significantly expressed in tumor tissue in TCGA database($p < 0.001$, Fig.

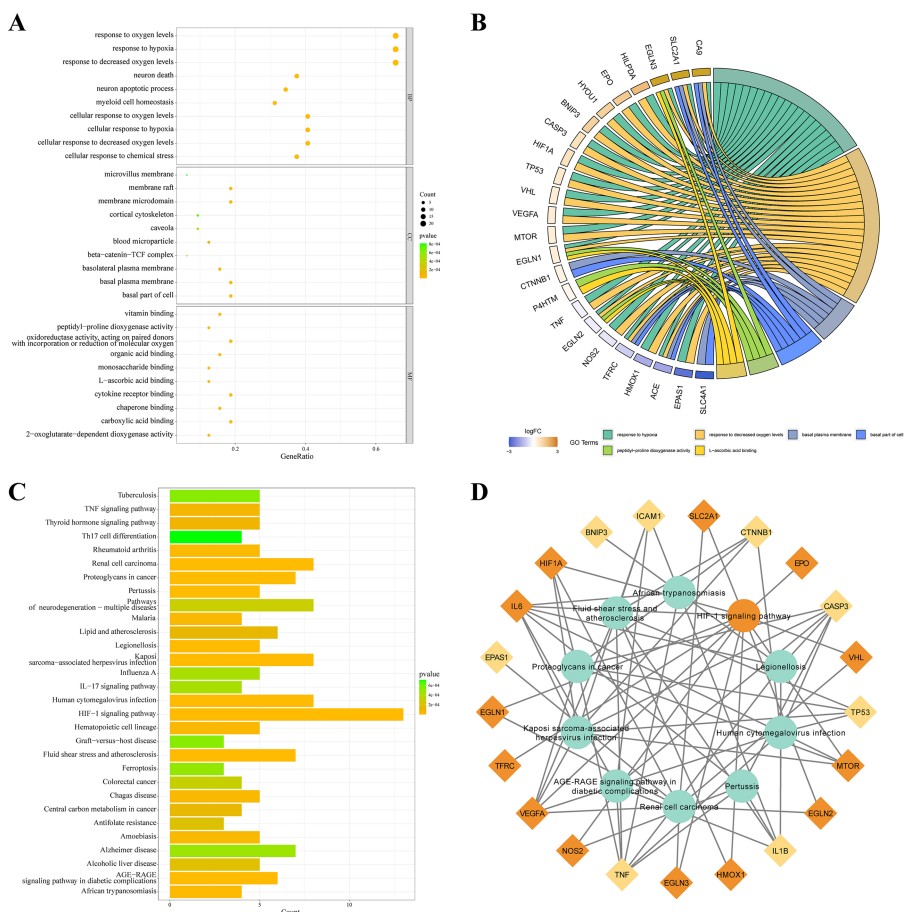

**Figure 3** **KEGG and GO analysis of 32 DEGs.** (A–B) GO enrichment of IHRGs in biological processes (BP), cellular component (CC), and molecular function (MF). (C–D) The KEGG analysis revealed the IHRGs.

8B). Meanwhile, the prognosis of the high-expression group of GSEC was worse than that of the low-expression group ($p < 0.001$, Fig. 8C). Further, a total of five intermittent hypoxia-related mRNAs (EGLN3, CASP3, TCF4, TFRC, and VHL) were identified as the downstream targets of GSEC from TargetScan databases (Fig. 8D). Utilizing the GEPIA database, we revealed that only EGLN3 was upregulated in lung cancer tissues compared to normal tissues (Fig. 8E). Thus, the ceRNA of lncRNA GSEC/miR-873-3p/EGLN3 axis was constructed (Fig. 8F).

## Expression of lncRNA GSEC, miR-873-3p, EGLN3 in LUAD

RT-qPCR analysis was used to detect the expression of lncRNA GSEC, miR-873-3p, and EGLN3 in LUAD. Expression of lncRNA GSEC and EGLN3 in LUAD cell lines was higher than in normal cell lines (Figs. 9A, 9C). Expression of miR-873-3p in LUAD cell lines was lower than in normal cell lines (Fig. 9B). Then, we found that the expression of EGLN3 in LUAD cell lines was higher than in normal cell lines at the protein level (Fig. 9D). Figure 9E indicated that the transfection was successful. Meanwhile, lncRNA GSEC

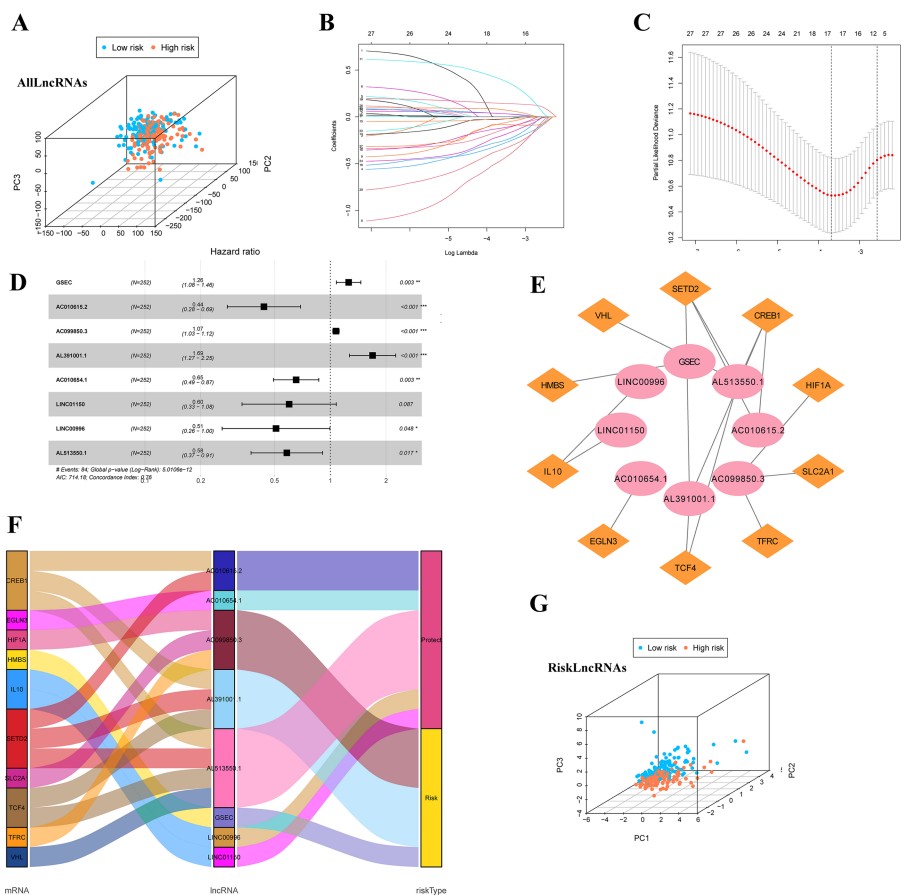

**Figure 4** **Identification of intermittent hypoxia-related lncRNAs (IHRLs) and their subsistence analysis.** (A) PCA of all LncRNAs. (B–C) The LASSO Cox algorithm was used to establish a prognosis model. (D) Eight lncRNAs were selected to construct the risk model. (E) Co-expression structure between IHRLs and genes. (E) Sankey diagram of the co-expression network. (F) The PCA visualized the different distribution patterns of patients in eight intermittent hypoxia-related lncRNAs.

interference inhibits the proliferation of A549 and PC9 cells (Figs. 9F, 9G). Similarly, migration functions were attenuated after interference with lncRNA GSEC (Figs. 9H, 9I). Finally, the silencing of GSEC inhibits the cloning capacity of A549 and PC9 cells (Figs. 9J, 9K).

## LncRNA GSEC sponges MiR-873-3p

To explore the potential molecular mechanisms of lncRNA GSEC in LUAD, RT-qPCR was performed following si-GSEC or miR-873-3p mimic transfection. The result showed that silencing of lncRNA GSEC promoted the expression of miR-873-3p compared to the 'si-ctrl' (Fig. 10A), and over-expression of miR-873-3p decreased the expression of GSEC Fig. 10B, Figure 10C analyzed binding sites between GSEC and miR-873-3p. Luciferase assay verified the relationship between GSEC and miR-873-3p (Figs. 10D–10E). RIP assay showed that GSEC and miR-873-3p performed immunoprecipitation in the Ago2 complex in A549 and PC9 cell lines (Figs. 10F–10G).

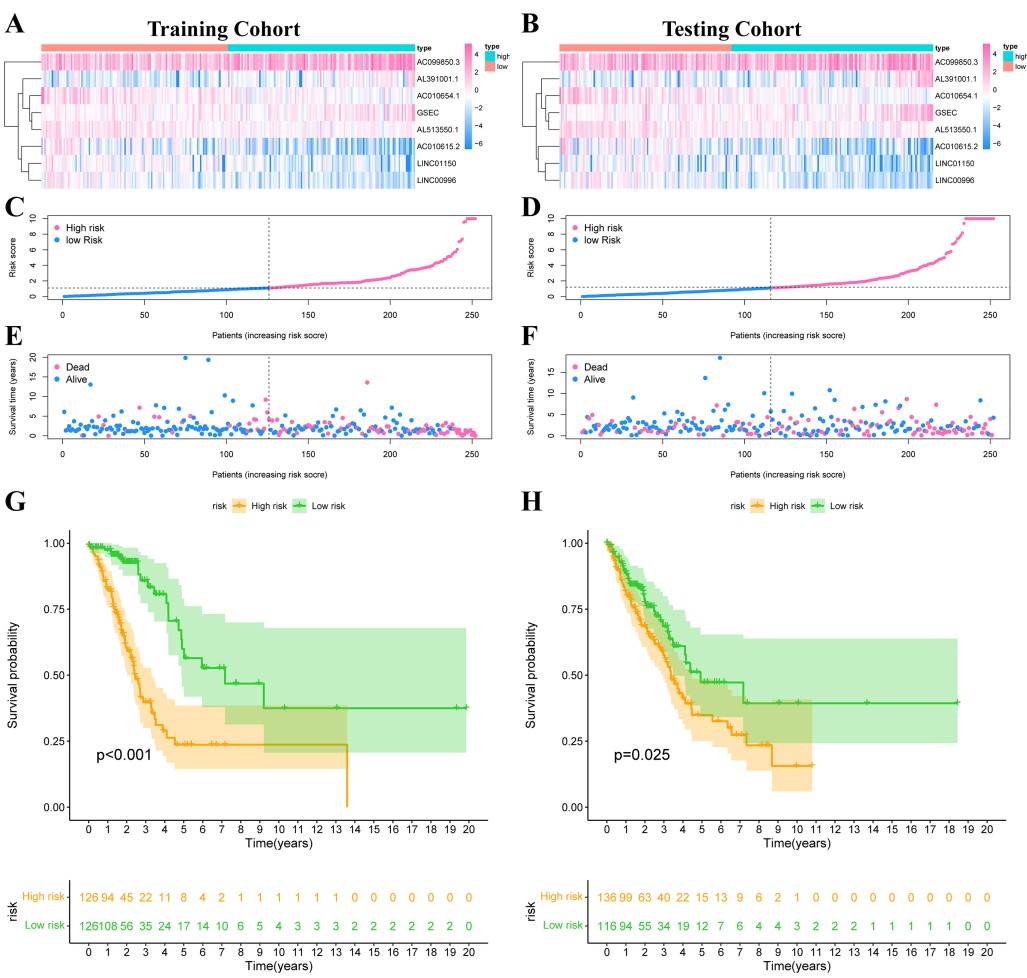

**Figure 5   Construction of risk model in LUAD patients.** (A–B) The heatmap of the expression of 8 IHRLs. (C–D) The risk score of LUAD patients in the two cohorts. (E–F) The survival status of LUAD patients in the two cohorts. (G–H) The Kaplan–Meier analysis of the OS of LUAD patients.

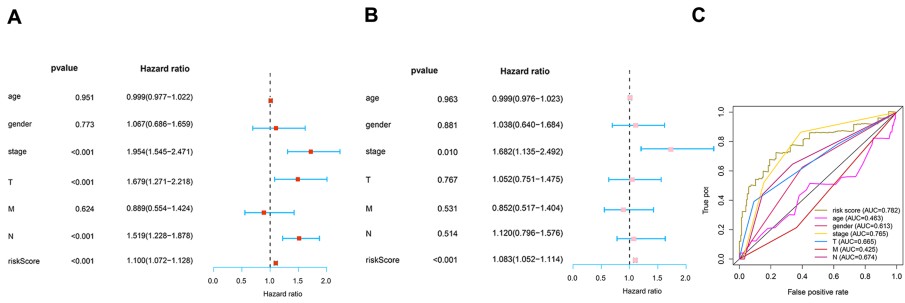

**Figure 6   Prognosis value of the risk model.** (A) Univariate analysis of clinical features. (B) Multivariate analysis of clinical features. (C) The ROC curves of the risk model (risk score: AUC =0.782).

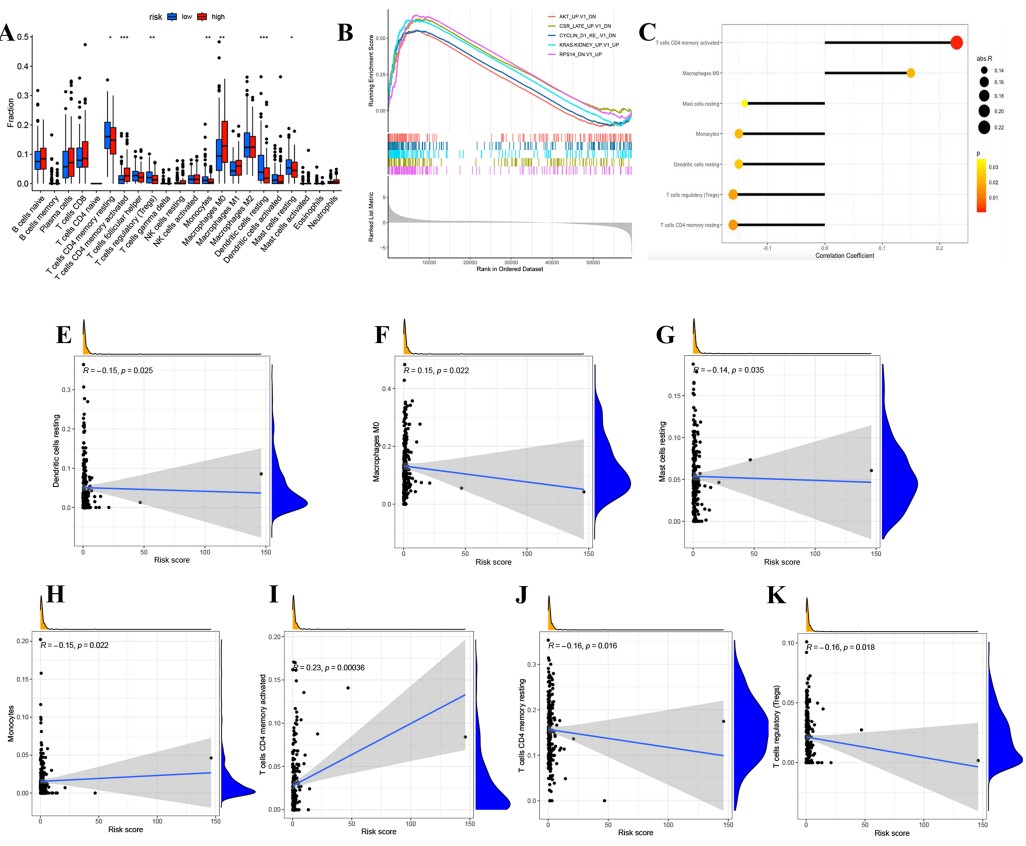

**Figure 7** **Analysis of immune activity.** (A) Comparison in immune cells of risk groups. (B) GSEA enrichment of oncogenic gene sets. (C-J) The relationship between infiltration of immune cells and the risk model. *$p < 0.05$; **$p < 0.01$; ***$p < 0.001$.

### Interference of miR-873-3p reverses the effect of lncRNA GSEC

To clarify whether lncRNA GSEC was affected by miR-873-3p, we co-transfected miR-873-3p inhibitor with si-GSEC. Fig. 11A indicated that the transfection was successful. The proliferation was restrained by si-GSEC and rescued by the miR-873-3p inhibitor (Figs. 11B–11C). Meanwhile, the migration was suppressed by si-GSEC but rescued by a miR-873-3p inhibitor (Figs. 11D–11E). Also, clone capacity was inhibited by si-GSEC but rescued by the miR-873-3p inhibitor (Figs. 11F–11G) Above all, GSEC interference inhibits malignant activities by upregulating miR-873-3p in LUAD.

### LncRNA GSEC sponges miR-873-3p to upregulate EGLN3

Figure 12A analyzed binding sites between miR-873-3p and EGLN3. Luciferase assay verified the relationship between miR-873-3p and EGLN3 (Figs. 12B–12C) Moreover, the expression of EGLN3 was inhibited by the over-expression of miR-873-3p at both mRNA (Fig. 12D) and protein levels (Figs. 12E–12F) Meanwhile, we investigated that lncRNA GSEC inhibitor inhibited the expression of EGLN3 but could be reversed by a miR-873-3p inhibitor (Fig. 12G), which was consistent with the result of the Western blot (Figs. 12H–12I).

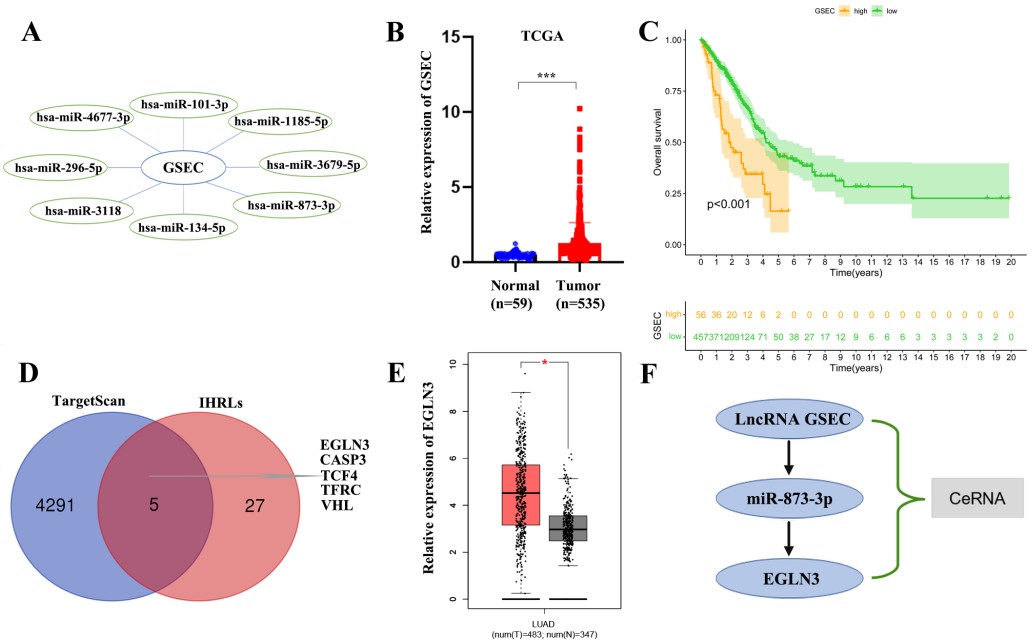

**Figure 8** **Construction of a ceRNA of lncRNA-miRNA-mRNA.** (A) LncRNA GSEC bound to eight miR-NAs. (B) The expression of lncRNA GSEC. (C) Kaplan–Meier survival of lncRNA GSEC. (D)Venn diagram identified the downstream targets in the TargetScan database and IHRLs. (E)The expression of EGLN3 in the GEPIA database. (F) The ceRNA of lncRNA GSEC/miR-873-3p/EGLN3 regulatory axis. $*p < 0.05$; $***p < 0.001$.

## DISCUSSION

Our research systematically identified IHRLs based on intermittent hypoxia-correlated mRNAs in LUAD. Current research shows intermittent hypoxia is of great significance in the occurrence and development of tumors (*Vilaseca et al., 2017*; *Chen et al., 2018*; *Li et al., 2021*). Scientists are also interested in this field of research. In various cancers, lncRNAs have been shown to play a crucial role in intermittent hypoxia, which has been confirmed by multiple research (*Chen et al., 2022*; *Hao et al., 2022*).

Therefore, we focused on the interaction of IHRLs with LUAD tumors. In this research, we screened 32 DEGs for variations in mRNA expression levels using the LUAD data of the TCGA database, which was then analyzed using bioinformatics methods. According to GO enrichment analysis, the DEGs were mainly associated with hypoxia, chemical stress, neuron death, carboxylic acid binding, oxidoreductase activity, and cytokine receptor binding. The investigations show intermittent hypoxia causes oxidative stress due to mitochondrial response inside mouse neuron death (*Douglas et al., 2010*). Pro-inflammatory cytokines play a role at several levels of hypoxic chemical reflex and cardiovascular control pathways, which may contribute to CIH-induced cardiorespiratory alterations (*Del Rio et al., 2012*). The above studies imply that intermittent hypoxia can result in multisystem illness. We utilized KEGG pathway analysis to identify that the DEGs were involved in the HIF-1 signalling pathway, AGE-RAGE signalling pathway, IL-17 signalling pathway, and

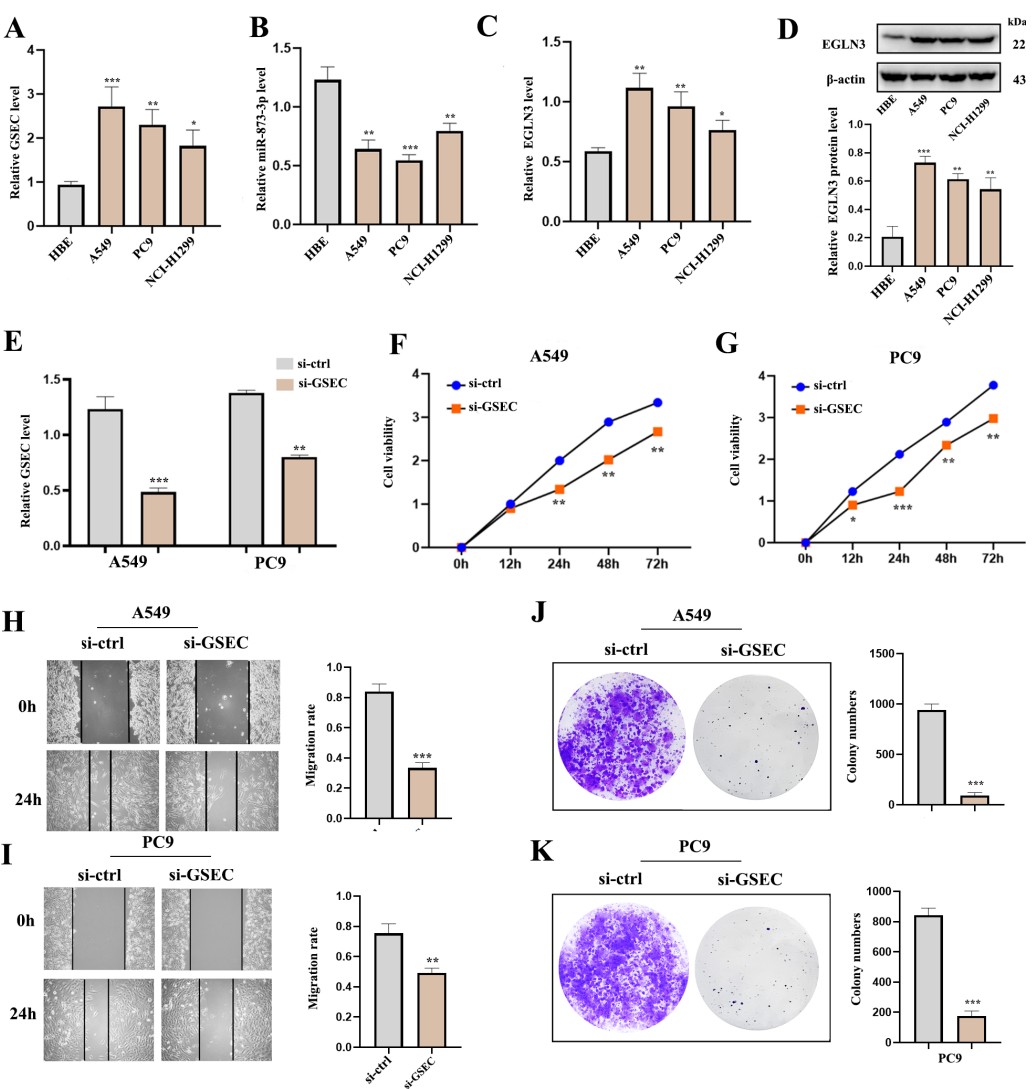

**Figure 9** **The expression and the effect of lncRNA GSEC, miR-873-3p, and EGLN3 in LUAD.** (A) Expression of GSEC in different LUAD cell lines compared to normal astrocytes. (B) Expression of miR-873-3p in different LUAD cell lines compared to normal astrocytes. (C) Expression of EGLN3 in different LUAD cell lines compared to normal astrocytes. (D) Expression of EGLN3 at the western blot level. (E) The efficiency of GSEC knockdown (si-GSEC). (F-G) The proliferation of cells by CCK-8 assays. (H-I) The migration of cells by wound healing assays. (J-K) The cloning capacity of cells. *$p < 0.05$; **$p < 0.01$; ***$p < 0.001$.

ferroptosis. HIF-1A is a crucial pathway in intermittent hypoxia. Recent studies suggest intermittent hypoxia may interfere with mtROS in lung cancer cells to play a carcinogenic role through the HIF-1 α/ATAD2 pathway (*Hao et al., 2022*). Ferroptosis is a recently discovered type of cell death, and the ferroptosis inhibitor Fer-1 has been shown to reduce intermittent hypoxia-induced lung injury in rats (*Shah, Shchepinov & Pratt, 2018*; *Chen et al., 2022*). Numerous studies have demonstrated that molecular pathways and processes are intimately associated with the formation and progression of cancers (*Alinejad et al.,*

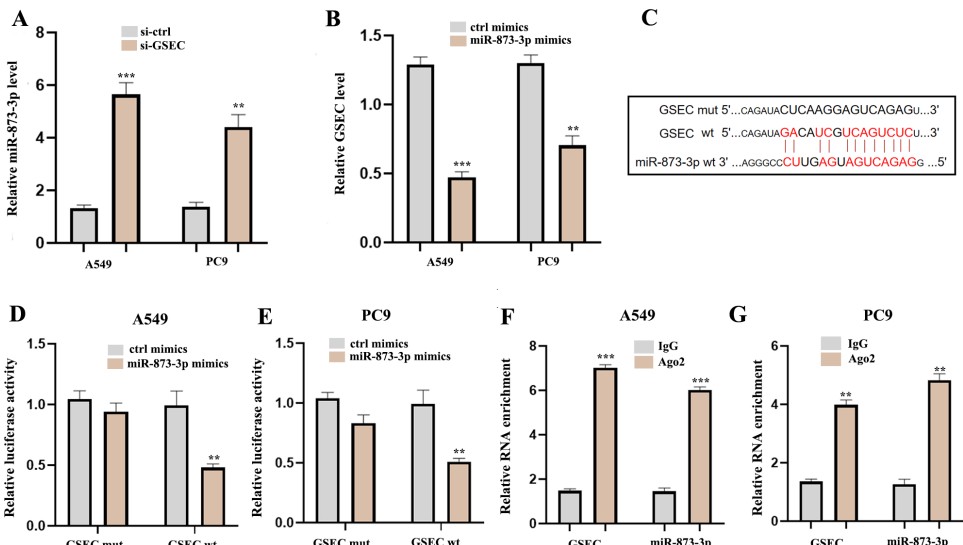

**Figure 10** **LncRNA GSEC acted as ceRNA for miR-873-3p.** (A) Expression of miR-873-3p following si-GSEC transfection. (B) Expression of GSEC following overexpression of miR-873-3p. (C) The predicted interacting sites. (D–E) The relationship between GSEC and miR-873-3p was performed by Dual-Luciferase reporter assay. (F–G) The immunoprecipitation of GSEC and miR-873-3p was examined by the RIP experiment. **$p < 0.01$; ***$p < 0.001$.

*2017*; *Waghela et al., 2021*; *Lei, Zhuang & Gan, 2022*; *Lin et al., 2022*). In addition, eight key IHRLs were identified after univariate Cox regression and Lasso-Cox regression analysis, and a prognostic model based on them was established. The combination of multivariate COX regression and univariate COX assists in identifying the LncRNA signature in lung cancer (*Mo et al., 2022*). Finally, the intermittent hypoxia-related prognostic lncRNA signature could effectively predict the survival risk of LUAD patients.

Numerous research has demonstrated that lncRNAs can function as competing endogenous RNAs and sponge microRNA (miRNA) sites (ceRNAs) (*Wang et al., 2019*; *Conte et al., 2021*; *Yang et al., 2021b*). Furthermore, investigations have revealed that lncRNAs associated with ceRNAs play a critical role in the occurrence and progression of malignancies (*Shi et al., 2021*; *Ye, Li & Zhao, 2021*; *Zhao, Feng & Tang, 2021*). lncRNA SNHG16 enhances lung cancer cell proliferation, migration, and invasion by modulating the miR-520/VEGF axis (*Chen et al., 2020*). LncRNA SPINT1-AS1 stimulates the growth of breast cancer cells through sponge let-7a/b/i-5p (*Zhou et al., 2021*). Through stabilizing GLUT1, the LncRNA GAL enhances colorectal cancer liver metastasis (*Li et al., 2022*). These results indicated that ceRNAs had been extensively studied and reported in various cancers (*Yang et al., 2021a*). According to prior investigations, we hypothesized a new axis of lncRNA GSEC/miR-873-3p/EGLN3 using bioinformatics approaches. Meanwhile, the axis was confirmed by experimentation. GSEC performs a vital function in tumor formation and progression.LncRNA GSEC has an essential role in the occurrence and progression of numerous forms of cancer. LncRNA GSEC promotes osteosarcoma's proliferation, migration, and invasion by targeting the miR-588/ EIF5A2 axis (*Liu et al., 2020*). LncRNA

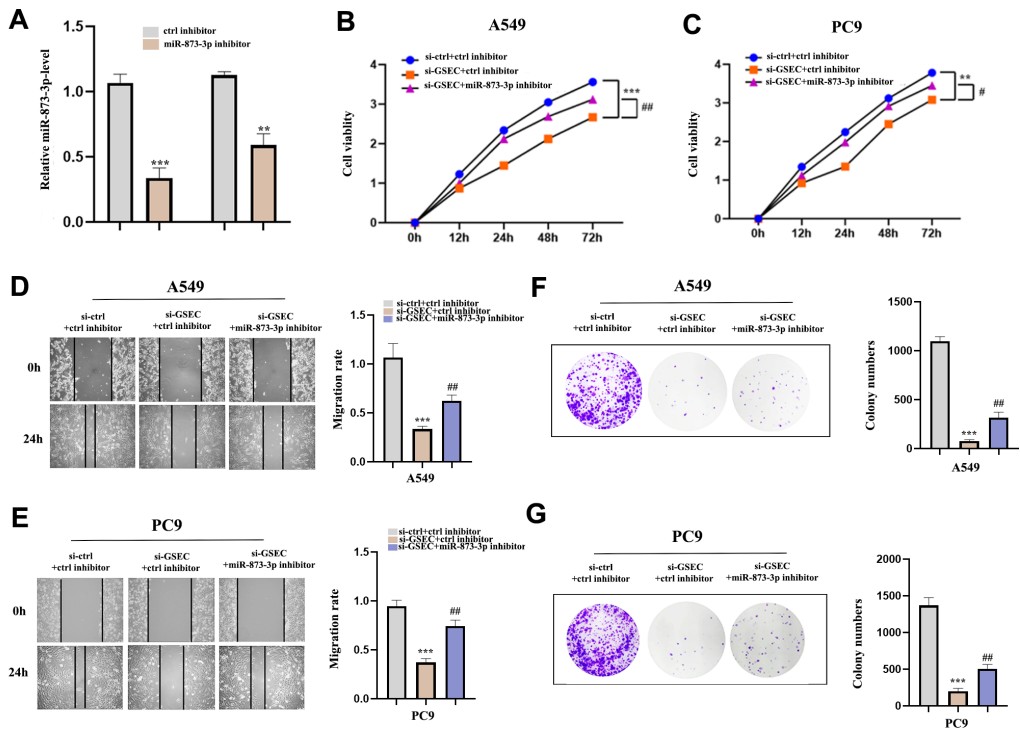

**Figure 11** **MiR-873-3p reversed the lncRNA GSEC knockdown effects on LUAD cells.** (A) The expression of miR-873-3p. (B–C) The proliferation of cells detected by CCK-8 assays. (D–E) The migration of cells detected by wound healing assays. (F–G) The cloning capacity of cells was identified by colony formation assay. $**p < 0.01$; $***p < 0.001$; *vs.* ctrl inhibitor; $^{\#}p < 0.05$; $^{\#\#}p < 0.01$ *vs.* si-GSEC+ctrl inhibitor.

GSEC promotes breast cancer progression by sponging the miR-202-5p/AXL Axis (*Zhang et al., 2021*). LncRNA GSEC encourages the progression of hepatocellular carcinoma by targeting the miR-101-3p/SNX16/PAPOLG axis (*Hu et al., 2022*). We observed that lncRNAs GSEC and EGLN3 were over-expressed in LUAD, and miR-873-3p was down-regulated, similar to previous results on lncRNA GSEC. Furthermore, *Cheng et al. (2019)* revealed that miR-873-3p expression was suppressed and acted as a tumor suppressor in LUAD. The above results confirmed our conclusions about the low word of miR-873-3p in LUAD.In our investigation, the knockdown of miR-873-3p attenuated the effect of lncRNA GSEC interference on A549 and PC9 cells. Our data revealed that miR-873-3p worked through the ceRNA regulatory mechanism. Utilizing the TargetScan database, the intermittent hypoxia-related mRNA (EGLN3) was identified as the downstream target of miR-873-3p. EGLN3 is associated with various tumor conditions, and USP9X relieves cholangiocarcinoma by upregulating EGLN3 (*Chen et al., 2021*; *Jin et al., 2022*).

Our research using RT-qPCR and western blotting methods revealed that EGLN3 was significantly more upregulated in A549 and PC9 cell lines than in the normal group.MiR-873-3p was a mediator in the regulation process between lncRNA GSEC and EGLN3. Consequently, we concluded that lncRNA GSEC acts as an oncogene in lung adenocarcinoma by targeting miR-873-3p to modulate EGLN3, which might be

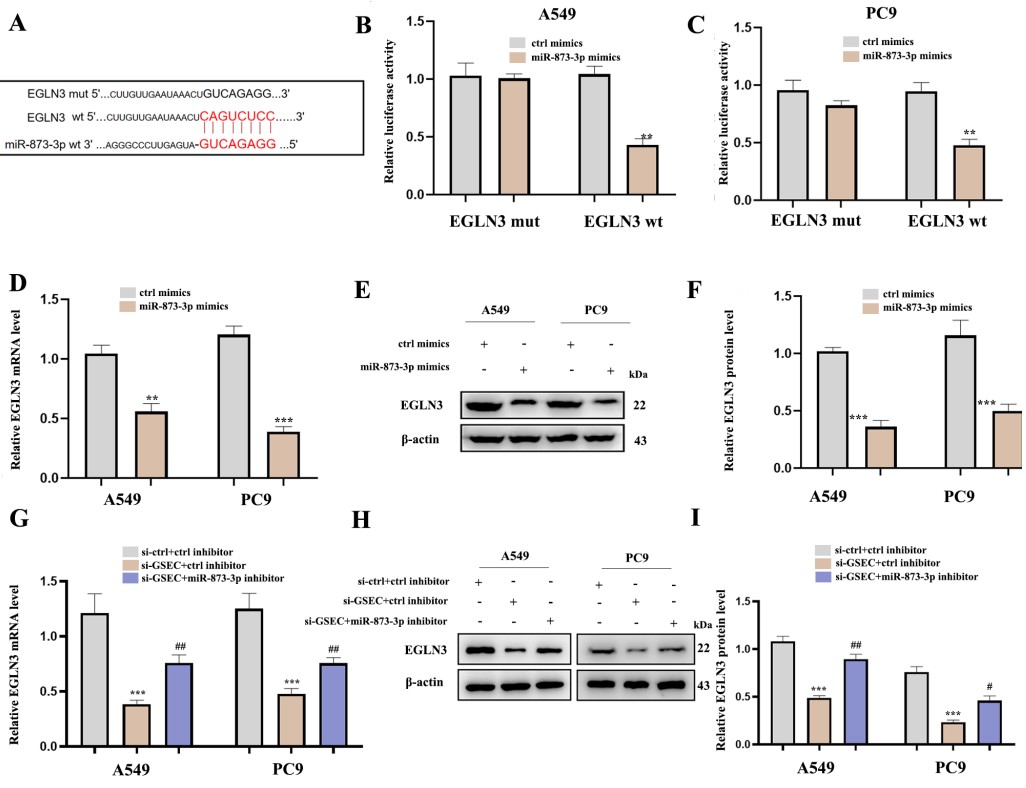

**Figure 12 EGLN3 was a downstream target of miR-873-3p and lncRNA GSEC sponged miR-873-3p to upregulate EGLN3.** (A) The predicted interacting sites. (B–C) The relationship between EGLN3 and miR-873-3p was performed by Dual-Luciferase reporter assay.** $p < 0.01$; *vs.* ctrl mimics. (D–F) Expression of EGLN3 following miR-873-3p overexpression. ** $p < 0.01$; *** $p < 0.001$; *vs.* ctrl mimics. (G–I) Expression of EGLN3 following si-GSEC or si-GSEC+miR-873-3p inhibitor. *** $p < 0.001$; *vs.* ctrl inhibitor; # $p < 0.05$; ## $p < 0.01$ *vs.* si-GSEC+ctrl inhibitor.

the mechanism of LUAD progression. This study strongly supports increasing evidence advocating for measuring tumor hypoxia before treatment to select patients most likely to benefit from hypoxia modification in lung cancer. Our research suggests that this polygenic test may help provide treatment options for patients with hypoxic lung cancer.

Unfortunately, there are still some limitations to our study. Our raw data are from publicly available databases. Using publicly available databases can be beneficial and limiting in research studies. While these databases provide a vast amount of data that can be accessed and analyzed, some potential limitations and biases could affect the findings. One limitation is the quality and accuracy of the data within these databases. Another potential limitation is the representativeness of the data in publicly available databases.

Furthermore, the data in publicly available databases may be subject to publication bias. It is also important to note that publicly available databases may not always be up-to-date or comprehensive. The application of mutual validation demonstrates the viability of this risk model, and more data may be needed for further verification. Meanwhile, we preliminarily

validated the ceRNA of intermittent hypoxia-related lncRNA GSEC/miR-873-3p/EGLN3 regulatory axis *in vitro*. More experiments need to be performed *in vivo*.

## CONCLUSIONS

Identifying a prognostic model related to intermittent hypoxia and specific lncRNAs provides a foundation for further investigation. The predictive risk model developed in this study can potentially enhance patient care and treatment decision-making. Furthermore, using this predictive risk model can have implications for healthcare policy-making. The findings of this study have a significant impact on future research, clinical practice, and policy-making in the field of LUAD. Further research can build upon these findings to deepen our understanding of LUAD progression and identify potential therapeutic targets. The predictive risk model can potentially improve clinical decision-making and patient outcomes while also informing healthcare policies to optimize resource allocation and patient care. Overall, this study has the potential to significantly impact the field of LUAD and contribute to advancements in patient care and treatment strategies.

### Funding
This research was supported by the National Natural Science Foundation of China (No. 81970082). The funders had no role in study design, data collection and analysis, decision to publish, or preparation of the manuscript.

### Grant Disclosures
The following grant information was disclosed by the authors:
National Natural Science Foundation of China: 81970082.

### Competing Interests
The authors declare there are no competing interests.

### Author Contributions

- Peijun Liu conceived and designed the experiments, performed the experiments, analyzed the data, prepared figures and/or tables, authored or reviewed drafts of the article, and approved the final draft.
- Long Zhou conceived and designed the experiments, performed the experiments, analyzed the data, prepared figures and/or tables, authored or reviewed drafts of the article, and approved the final draft.
- Hao Chen performed the experiments, prepared figures and/or tables, and approved the final draft.
- Yang He performed the experiments, prepared figures and/or tables, authored or reviewed drafts of the article, and approved the final draft.
- Guangcai Li conceived and designed the experiments, analyzed the data, authored or reviewed drafts of the article, and approved the final draft.

- Ke Hu conceived and designed the experiments, analyzed the data, authored or reviewed drafts of the article, and approved the final draft.

## Data Availability

The raw measurements are available in the Supplemental Files.

## Supplemental Information

Supplemental information for this article can be found online at http://dx.doi.org/10.7717/peerj.16242#supplemental-information.

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
