# Peer review of "Identification of a novel intermittent hypoxia-related prognostic lncRNA signature and the ceRNA of lncRNA GSEC/miR-873-3p/EGLN3 regulatory axis in lung adenocarcinoma"

_PeerJ, doi:10.7717/peerj.16242_

## Round 0.1 · original submission · Major Revisions

Please revise the manuscript as suggested.

·

Basic reporting

The research demonstrated that the results of the present study demonstrated that the hypoxia-related prognostic lncRNA risk signature had the potential to predict the prognosis of cancer. However there are some points to improve.
1. A lot of similar articles used the LASSO regression model to establish the prognosis prediction model for lung adenocarcinoma which might not be possible under current medical conditions since the prognosis is not simply influenced by several genes expression. The authors should use advanced artificial intelligence models rather than traditional machine learning models such as Random Forest or LASSO models.
2.The bioinformatics analysis of this article is not innovative and in-depth, many of which are online website analysis data.
3.The methods section is not detailed enough, and the statistical analysis section lacks.
4.The discussion in this study is not deep enough, and it is suggested that it be discussed in depth.
5.It is recommended that the language of this paper be embellished.
6. Why did the author divide patients into training and validation cohorts with a ratio of 1:1, why not 7:3 or 6:4. In predictive model construction, 6:4 or 7:3 are the most common ratio used to cohort development. If there are too many patients in the training cohort, you could not guarantee the generalbility of the model. On the contrary, if there are too many patients in the validation cohort, the model won't be fitted well.

Experimental design

no comments

Validity of the findings

no comments

Additional comments

no comments

Reviewer 2 ·

Basic reporting

The research presented is insightful, however, I have noticed a few areas if improved will enhance the readability of the manuscript.
• Grammatical Errors and Sentence Structure: There are several instances where sentences could be restructured for clarity. For example, in Line 72-73, the phrase "in the previous" could be revised to "in previous studies" for better understanding.
• Spacing Inconsistencies: Please ensure consistent spacing throughout the manuscript. For instance, in Line 67-68, there should be a space after the period and before the reference "(Shobatake et al.,2022)". Similarly, in Line 149, there is an unusually large space between "37" and "C" that should be corrected.
• Margin Alignment: Please ensure that the margins are aligned for all sentences. In Lines 101-103, the word "clinical" is split between two lines. This could be corrected for a better reading experience.
• Use of Acronyms: To make the manuscript more accessible, please provide the full forms of acronyms at their first mention.
These are just a few examples, and I recommend a thorough proofreading of the entire document to correct such issues and ensure clear and precise language.

Experimental design

The research is scientifically important as it explores the role of lncRNAs in LUAD, a prevalent type of respiratory cancer. By identifying a novel intermittent hypoxia-related prognostic lncRNA signature and the ceRNA of lncRNA GSEC/miR-873-3p/EGLN3 regulatory axis, the study provides valuable insights that could potentially lead to new diagnostic and therapeutic targets for LUAD.

However, the methodology section could be expanded to provide more details about the experimental procedures. The provide information is not sufficient at the moment to replicate the study.

Validity of the findings

The findings of this study are relevant not only to researchers in the field of oncology but also to clinicians treating patients with LUAD. The identification of a prognostic model related to intermittent hypoxia could be a valuable tool in assessing patient prognosis and guiding treatment decisions. Furthermore, the identification of potential therapeutic targets could pave the way for the development of new treatments for LUAD.

However, here are some suggestions to improve the paper:

1) Provide more context in the introduction: The introduction could provide more background information about the role of lncRNAs and intermittent hypoxia in lung adenocarcinoma. This would help the reader understand the significance of the study and its contribution to the existing body of knowledge.

2) While the limitations section does mention a couple of significant constraints, it would be beneficial to discuss them in more detail. For example, how does the use of publicly available databases potentially limit the findings? Are there known biases or errors in these databases that could affect the results?

3) In the conclusion, discuss the potential implications of your findings for future research, clinical practice, and policy-making. This will help to highlight the significance of the study and its potential impact.

Reviewer 3 ·

Basic reporting

no comment

Experimental design

no comment

Validity of the findings

no comment

Additional comments

This study is a bioinformatic study to evaluate hypoxia-related long noncoding RNA and tumor immune microenvironment (TIME) in lung adenocarcinoma This is a very timely and interesting research since novel gene signatures to clarify the heterogeneity of TME in LUAD, although there are some problems to be resolved.
1.The processing of the original data is not mentioned, such as the format of the downloaded data (count,FPKM,TPM?)
2.. What is Chemotherapy of the high and low risk groups.
3.The authors should elaborate the difference between the current manuscript and previous similar published papers, and explain the advantages of current study.
4.The discussion should be expanded and the authors should speculate more regarding the impact of this study on clinical practice, especially in terms of translational value.
5.My biggest concern lies in the random division of the traning set and testing set. Since the training set and testing set were randomly divided, I would imagine that the constructed prognostic signature are highly dependent on the training set and would vary with different sample divisions. However, the authors did not demonstrate any analysis with respect to it.
6.There are many methods to evaluate immune cell infiltration, such as MCPcounter, TIMER, etc. Why only CIBERSORT is used in this paper?
7.How to deal with missing values of some clinical features in TCGA data and data screening criteria. Different processing criteria will get different results.
8.The model building method in this paper is very common and not novel. Why not use principal component analysis to build the model.

---

## Round 0.2 · accepted · Accept

This manuscript can be accepted now.

·

Basic reporting

no comment

Experimental design

no comment

Validity of the findings

no comment

Additional comments

no comment

Reviewer 2 ·

Basic reporting

No comments

Experimental design

No comments

Validity of the findings

No comments

Additional comments

Dear Authors,

I've had an opportunity to review your revised manuscript and I must say, I'm quite impressed with the thoroughness of your revisions. Given the substantial improvements you've made, and considering the significant contribution this research brings to our field, I am in full support of this manuscript's publication.

Thank you for taking my feedback into account and for your diligence in making the necessary revisions. I eagerly anticipate seeing your valuable work published and am confident it will have a positive impact in our scientific community. Best of luck!

Reviewer 3 ·

Basic reporting

The author's revisions have greatly improved the manuscript.

Experimental design

No additional comments

Validity of the findings

No additional comments

Additional comments

No additional comments